# Early Prediction of Students' Performance Using a Deep Neural Network Based on Online Learning Activity Sequence

**Xiao Wen** [1,*]  **and Hu Juan** [2]

1   School of Educational Science, Anhui Normal University, Wuhu 241002, China
2   School of Computer and Software, Anhui Institute of Information Technology, Wuhu 241002, China; juanhu3@iflytek.com
*   Correspondence: xiaowen@ahnu.edu.cn

**Abstract:** Predicting students' performance is one of the most important issues in educational data mining. In this study, a method for representing students' partial sequence of learning activities is proposed, and an early prediction model of students' performance is designed based on a deep neural network. This model uses a pre-trained autoencoder to extract latent features from the sequence in order to make predictions. The experimental results show that: (1) compared with demographic features and assessment scores, 20% and wholly online learning activity sequences can achieve a classifier accuracy of 0.5 and 0.84, respectively, which can be used for an early prediction of students' performance; (2) the proposed autoencoder can extract latent features from the original sequence effectively, and the accuracy of the prediction can be improved more than 30% by using latent features; (3) after using distance-based oversampling on the imbalanced training datasets, the end-to-end prediction model achieves an accuracy of more than 80% and has a better performance for non-major academic grades.

**Keywords:** predict students' performance; deep neural networks; online learning activity sequence



## 1. Introduction

With the rapid development of information technology, online learning systems such as Learning Management Systems (LMSs) and Massive Open Online Courses (MOOCs) have been widely used to support learners' self-regulated and hybrid learning. These systems not only provide learners with learning resources such as videos, documents, questionnaires, and quizzes, but also trace learners' learning activities and generate records of their learning behavior. They can help instructors identify characteristics of learning behavior and predict students' performance through an in-depth analysis of these records. At present, many researchers use methods such as data mining and machine learning to extract valuable information from these records, leading to the emergence of educational data mining [1], an interdisciplinary research field.

Predicting students' performance is one of the most important issues in educational data mining, and researchers have made remarkable achievements regarding this issue. By predicting students' performance, instructors can understand the situation of learners effectively and guide learners who may be at risk of academic failure as soon as possible. Researchers collect records including learners' demographic features, historical academic performance and learning behavior, and then establish prediction models using decision trees, Naive Bayes, logistic regression, K-Nearest Neighbor (KNN) and other algorithms to predict students' performance [2]. Due to these systems' capacity to automatically track learners' online learning activities, increasingly, researchers have started to extract valuable information from large-scale, high-quality learning behavior records. In recent years, deep neural networks have achieved great success in many fields [3], such as image recognition, natural language processing, and anomaly detection. Increasingly, researchers have begun to use deep neural networks to solve problems in educational data mining [4].

Luo et al. created 20 variables from the logs collected in an LMS to describe learners' online learning behavior, including login times, total course access and discussion posts, and they established a prediction model for students' performance using the Random Forest (RF) method [5]. Lee et al. collected the records of learners watching videos from an MOOC and established a regression model regarding the scores obtained on courses using a Feedforward Neural Network (FNN) [6]. Xie et al. gathered logs from three courses from SPOC, established dozens of variables to describe online learning behavior, and used ECOC-based algorithms to establish a prediction model of students' performance [7]. Although these predictions achieved high accuracy, there are two flaws. Firstly, the variables/features of learning behavior used to predict students' performance can only be created after the learners finish all their learning activities; therefore, it is impossible to predict performance in advance. Secondly, the established prediction models cannot extract more representative latent features from the raw records, which affects the accuracy of the predictions.

In this study, we proposed a representation of an online learning activity sequence, which can be used as the input for a deep neural network, so that the students' performance can be predicted at the beginning of and during online learning. In addition, we designed an autoencoder based on a deep neural network, which can extract latent features from the original representation to improve the accuracy of the predictions. Based on the representation and the autoencoder, we designed an end-to-end prediction model of a student's performance based on their individual online learning activity sequence.

Specifically, the contributions of this paper are as follows:

(1) We propose a representation of an online learning activity sequence that can be used as the input for a deep neural network to predict students' academic performance.

(2) We design an unsupervised autoencoder based on a deep neural network. This can extract latent features from the sequence of students' online learning activities, which can then be used for visual analysis and further improve the accuracy of our predictions.

(3) We designed an end-to-end prediction model of student's performance based on the online learning activity sequence. This model is composed of an autoencoder and a classifier based on a deep neural network. Aiming to address the imbalance in the training dataset, we use an algorithm based on K-Means and SMOTE to resample the training dataset, which improves the accuracy of the prediction model for non-major classes.

The rest of this article is organized as follows. In Section 2, we introduce the preliminaries and background of this study and discuss related works. We propose a new representation, autoencoder, and end-to-end prediction model in Section 3. The results of the experiment are shown and discussed in Section 4. In Section 5, we list the key findings of this study and discuss potential future research directions.

## 2. Problem Definition and Related Work

### 2.1. Problem Definition

In educational data mining, the prediction of students' performance is regarded as a classification task of supervised learning. Researchers collect a training dataset containing students' demographic features, assessment scores, online learning behavior variables, and corresponding performance to train the models established by algorithms based on machine learning. After training, the prediction results of a student's performance can be obtained by inputting the new student's features into the prediction model. A training dataset $D$ containing $m$ students can be expressed as $D = \{(S_1, P_1), (S_2, P_2), \ldots, (S_m, P_m)\}$; $S_i$ is the feature vector of the $i$-th student; and $P_i$ is the academic performance of the $i$-th student. The prediction model obtained by training can be regarded as a function $f$. For a student with an unknown performance of feature vector $S_x$, we can obtain the prediction result of performance $P_x$ by using $f$ based on $S_x$, which is then expressed as $P_x = f(S_x)$. If $S_x$ is a sequence of features, an early prediction means that using the part of $S_x$ in front can predict a student's performance.

### 2.2. Related Work

Online learning behavior analysis has become one of the most concerning issues in learning analysis and educational data mining. Researchers identify learning behavior patterns [8–11], measure learning engagement [12–14], identify cognitive styles [15,16], set up students' portraits [17,18] and recommend learning resources [13,19] by analyzing student records of online learning behavior. The results of these studies reflect that the records of students' online learning behavior contain a lot of important information in the learning process.

Luo et al. extracted 20 features from the logs of an LMS to describe learners' online learning behavior. These features were statistics of students accessing different online learning activities. They also used the Random Forest method to establish a prediction model of students' performance, expressed in six grades, A-F, with an average prediction accuracy of 49% in five types of courses [5]. Lee et al. gathered the logs of students' watching videos in two courses from MOOC, extracted 16 features of watching videos and 8 features of answering questions, and established a regression model of course scores using a feedforward neural network. Under the condition that the model contained eight hidden layers, the Mean Absolute Error (MAE) between the prediction results and actual scores was reduced to 6.8 [6]. Xie et al. collected logs regarding the mixed teaching of information technology courses in three universities from SPOC, but did not specify what features were created to describe online learning behavior. The experimental results showed that the prediction model established using the ECOC algorithm had a higher prediction accuracy than the classical SVM method, with a prediction accuracy of 80% [7]. Wang et al. collected logs of 488 undergraduates in 10 courses from Moodle, which is the most famous LMS, and extracted 10 indicators of activity engagement, including counting-based and duration-based features. They used the Partial Least Squares (PLS) method to establish a structural model of online behavioral engagement and learning achievement and analyzed the online engagement activities that have the greatest impact on learning achievement [20]. They extracted eight counting-based behavioral features from the logs and established a four-class prediction model using the decision tree, achieving a prediction accuracy of 70%. Because the decision tree is a typical white box model, they also extracted the most important six rules that affected academic performance from the established prediction model [21]. Zhang et al. extracted 19 behavioral features in the five stages of preparation, progress, resource learning, forum interaction, and test from the logs, and established a prediction model using logistic regression, achieving a prediction accuracy of 95% [22]. Chen et al. paid attention to the performance of online short-term courses, used the content features and behavioral features discussed in forums as indicators, and established prediction models using four classifiers: K-Nearest Neighbor (KNN), Support Vector Machine (SVM), Linear Discriminant Analysis (LDA), and Random Forest (RF). The experimental results showed that the performance of classifiers in different courses was not consistent, and the highest prediction accuracy was 89% [23].

Li et al. constructed a studen–tproblem interactive network using the mouse logs of students in the interactive online question library and used a Graph Neural Network (GNN) to predict students' performance, achieving an accuracy of 66% and 55% in short-term and long-term predictions, respectively. The prediction accuracy of their network was higher than the three classic classifiers of Gradient Boosting Decision Tree (GBDT), SVM and Logistic Regression (LR) [24]. Thomas et al. extracted the learning behavioral features from the logs of an LMS and trained a bidirectional Long Short-Term Memory (LSTM) Recurrent Neural Network, which achieved an 80% prediction accuracy [25]. Zhang et al. established a prediction model using multi-source sparse attention convolutional neural networks called MsaCNN to predict students' course grades according to the features of students and courses. The prediction accuracy was up to 84.9%, and the experimental results showed that the performance of this model was better than that of LR, SVM, KNN, decision tree, RF and other models [26]. Seyhmus et al. collected 3518 undergraduate students' logs from an LMS, extracted eight counting-based learning behavioral features, and established a

prediction model based on FNN to predict students' course grades, achieving a prediction accuracy of 80.47% [27]. Hajra et al. constructed the behavioral characteristics of learners based on VLE click streams, counted the numbers of clicks on different objects by learners in different time periods, and constructed an artificial neural network to predict learners' performance [28]. Some researchers use convolution, long-short-term memory (LSTM), and attention to aggregate sequences extracted from click logs to construct learners' behavior features [29–31]. However, these features still depend on learners' long-term logs, which cannot be used for early prediction. A comparison of related works is shown in Table 1.

**Table 1.** Comparison of related works.

| Ref. | Data Sources | Features | Machine Learning Model | Evaluation |
|---|---|---|---|---|
| [5] | Blended courses | 21 features of learning behavior | RF | Accuracy: 0.49 |
| [6] | MOOC | 16 features of watching videos<br>8 features of answering questions | FNN | MAE: 6.8 |
| [7] | SPOC | Not specifically indicated | ECOC | Accuracy: 0.8 |
| [21] | MOOC | 8 features of learning behavior | DT | Accuracy: 0.7 |
| [22] | Blended courses | 19 features of learning behavior | LR | Accuracy: 0.95 |
| [23] | Online short course | 3 features of content<br>10 features of learning behavior<br>4 features of students | SVM | Accuracy: 0.89<br>AUC: 0.8 |
| [24] | Online question library | 6 features of questions<br>12 features of mouse movement | GNN | Accuracy: 0.66 |
| [25] | LMS | 5 features of students<br>4 features of assign and exam | BiLSTM | Accuracy: 0.8 |
| [26] | LMS | Sequence of student grade records | CNN with attention | Accuracy: 0.85 |
| [27] | LMS | 8 features of learning behavior | FNN | Accuracy: 0.8 |
| [28] | OULA | 54 features of learning behavior | FNN with SVD | Accuracy: 0.86 |

In general, the results of related works indicate that students' online learning behavior records contain a lot of valuable information, such as students' learning style, degree of engagement and cognitive characteristics, and they can be used to predict students' performance. Deep neural networks perform better than the classical classifiers in predicting students' performance [32]. However, the features of learning behavior used in previous studies are based on statistics rather than sequence, and cannot be used to predict students' performance at the beginning of or during the learning process. Classical classifiers cannot extract more representative hidden features from the original data. In addition, the comparison of the features of online learning activities and other types of features used to predict students' performance is absent in previous studies.

## 3. Method

### 3.1. Representation of an Online Learning Activity Sequence

In this section, we propose a representation of the learning activity sequence, and the results can be used as the input for a deep neural network. This method can be used to represent the whole or partial sequence of online learning activities of learners, which is beneficial for predicting learners' performance as early as possible, rather than generating statistical features of online learning behavior only after the online course ends. There are $m$ students $S = \{s_1, s_2, \ldots, s_m\}$. The online learning process is divided into $n$ phases according to durations such as hours, days, or weeks, or according to different modules of the course $P = \{p_1, p_2, \ldots, p_n\}$. A collection of $c$ online learning activities that we observe is $A = \{a_1, a_2, \ldots, a_c\}$. We represent the sequence of learning activities of all students as a three-dimensional tensor $R^{m \times n \times c}$, and element $r_{i,j,k}$ in $R$ represents the weight of learning activity $a_k$ attended by student $s_i$ in phase $p_j$. The weight can be the time spent, the number of clicks or the scores obtained by a student on this learning activity recorded by the online learning system.

We use an example to explain this representation in detail. Table 2 contains some sample logs of students' learning behaviors recorded by the learning management system, including the duration of two students' three learning activities in two days. We take the duration directly as the weight of this activity. The sequence of these students' online learning activities can be represented by a three-dimensional tensor, as shown in Figure 1a.

**Table 2.** Sample logs of students' learning behaviors.

| Student | Date | Activity | Duration (Minute) |
|---|---|---|---|
| S1 | 1 January 2022 | A1 | 25 |
| S1 | 1 January 2022 | A2 | 30 |
| S1 | 1 January 2022 | A3 | 50 |
| S1 | 2 January 2022 | A1 | 10 |
| S1 | 2 January 2022 | A3 | 50 |
| S2 | 1 January 2022 | A1 | 20 |
| S2 | 1 January 2022 | A2 | 5 |
| S2 | 1 January 2022 | A3 | 100 |

```
[
    [25,30,50][10,0,50]
    [20,5,100][0,0,0]
]
```

```
[
    [1,1,0.5][0.4,0,0.5]
    [0.8,0.17,1][0,0,0]
]
```

(**a**) Original representation of activity sequences.    (**b**) Normalized representation of activity sequences.

**Figure 1.** Three-dimensional tensor representing the sequence of students' online learning activities.

As can be seen from Figure 1a, the representation of a student's learning activities in a phase is similar to the representation of Bag Of Word (BOW) [33], which is based on the assumption that there is no significant difference between different students' activity sequences in a phase. For the activities in which students do not participate in a phase, the representation sets the weight of these activities as 0. In order to avoid the influence of different orders of magnitude on the weights and accelerate the convergence of the deep neural network during the training, we use the Max–Min Value (1) to normalize all weights according to the type of activity. In this example, the weight of activity A1 is {25, 10, 20}, and the weight of A2 is {30, 5}, while the weight of A3 is {50, 50, 100}. After normalization, the final representation is shown in Figure 1b.

$$X' = \frac{X - X_{min}}{X_{max} - X_{min}} \tag{1}$$

### 3.2. Autoencoder of Learning Activity Sequence

Autoencoder is a self -supervised deep learning method, which has the ability to represent learning and dimensionality reductions. It has achieved great success in many fields, such as image compression and anomaly detection [34]. Using the proposed representation, the online learning behavior sequence of all students can be used as the input for the deep neural network, but this representation will lead to the high-dimensional curse [35]. If we want to use this sequence for prediction, we need to use a one-dimensional vector with $n \times c$ dimension as a student's feature. Usually, an online course is divided into many phases, according to days or weeks, and includes dozens of types of activities, so the dimensions of the feature vectors may be in the hundreds or even thousands, which seriously affects the accuracy of recognition and predictionn. To alleviate this problem, we

designed an autoencoder based on a deep neural network to reduce the dimension of the original feature vectors. The autoencoder can extract low-dimensional latent features from the original feature vectors and can be integrated with other deep neural networks as part of an end-to-end system. The structure of our autoencoder is shown in Figure 2.

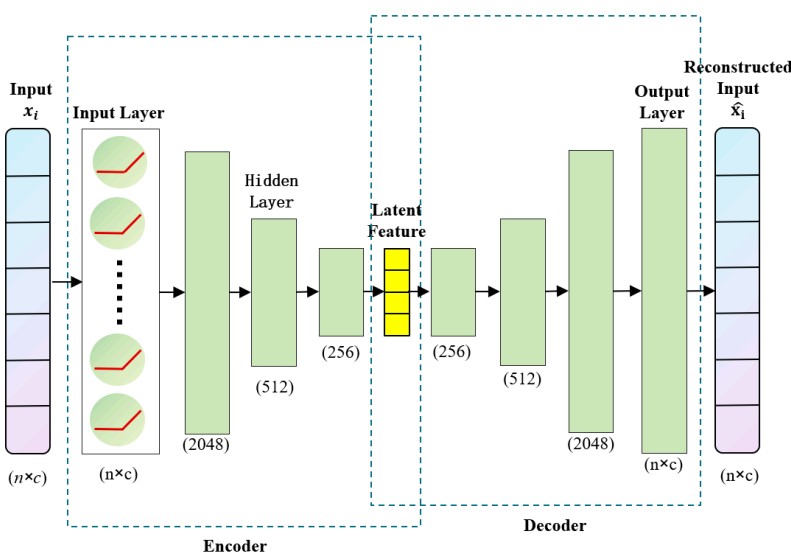

**Figure 2.** The structure of the autoencoder designed in this study.

As shown in Figure 2, the autoencoder we designed includes an input layer, an output layer, and six hidden layers. The number of neurons contained in the input layer and the output layer is equal to the dimension of the original feature vectors, and both are $n \times c$. The number of artificial neurons contained in other hidden layers is shown in this figure. The activation function used by all layers except the two hidden layers is the Rectified Linear Activation Function (RELU) [36], and the size of the latent feature generated by the encoder can be customized. During training, the autoencoder uses the cross-entropy function (2) to obtain the loss between the original input and the reconstructed input.

$$loss = -\frac{1}{m}\sum_{i=1}^{m}[x_i ln \hat{x}_i + (1 - \hat{x}_i)\ln(1 - x_i)] \tag{2}$$

After pre-training, the autoencoder can be integrated with other classifiers to form a new prediction model. In the training process, the autoencoder requires three parameters: the number of epochs, named countepoch, the size of a batch, named batch_size, and an optimizer for updating the parameters of the artificial neurons, named optimizer. For each batch of the training dataset, we need to obtain the loss between the output of the autoencoder and the original input and use the optimizer to update the parameters of all neurons in the autoencoder according to the gradients.

### 3.3. End-to-End Prediction Model of Students' Performance

Based on the method of representation and the use of the autoencoder to extract latent features from online learning activity sequences, as described in the previous two subsections, we designed an end-to-end performance prediction model based on each student's online learning activity sequence. The structure of this prediction model is shown in Figure 3. This model can take part of the learner's online learning activity sequence as input, and use the pre-trained autoencoder to extract latent features from the sequence to predict the learner's performance as soon as possible.

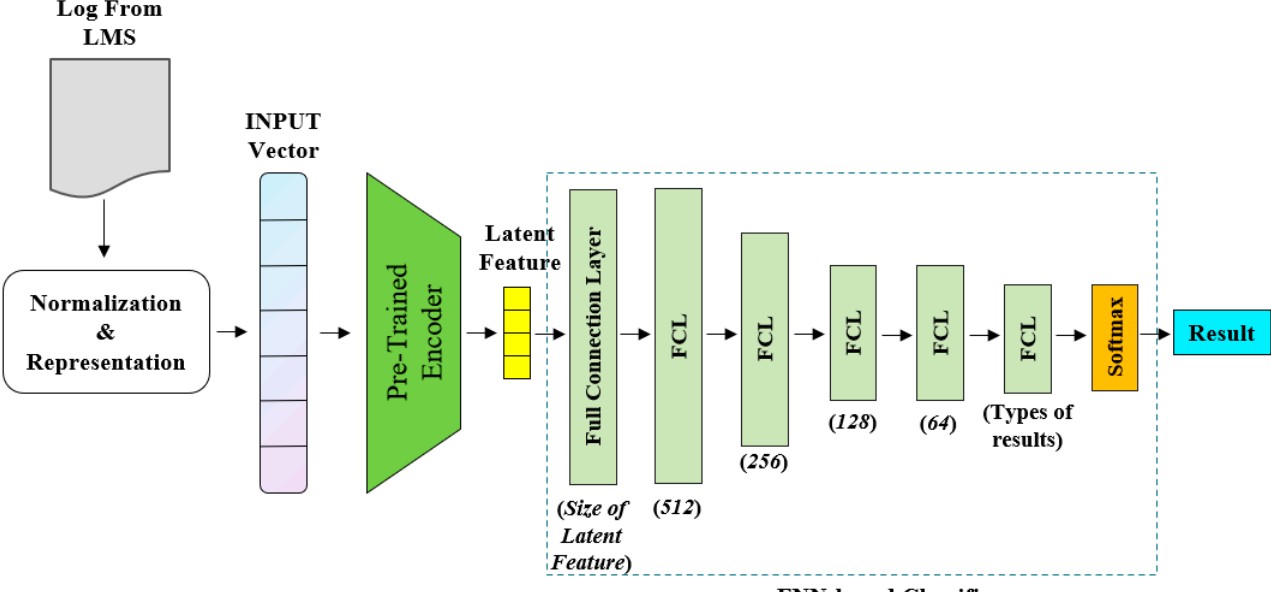

**Figure 3.** The structure of the proposed end-to-end prediction model of students' performance.

As can be seen from Figure 3, the input for this model is the students' logs of learning behavior, obtained from an LMS. Logs are normalized and represented to generate the input for the autoencoder. The autoencoder extracts the latent features from the original feature vectors for the FNN-based classifier to predict students' performance. The number of neurons contained in the last hidden layer is equal to the number of prediction grades. The result of the prediction model is the students' grades.

The hyperparameters' setting is one of the most important factors affecting the performance of a deep neural network [37]. The FNN-based classifier we designed contains five hidden layers, and the maximum number of neurons in a hidden layer is 512. These two important hyperparameters come from our previous survey. The survey results show that when the number of hidden layers exceeds five and the maximum number of neurons in the hidden layer is 512, FNN can generally achieve a prediction accuracy of more than 80% in various tasks predicting students' performance.

The training classifier and training autoencoder are supervised learning [38] and self-supervised learning [39], respectively, which is the most significant difference between them. The parameters required for training a classifier named countepoch, batch_size, optimizer are the same as those in the training autoencoder. We oversampled the training dataset at the beginning. This is because almost all training datasets on students' performance are imbalanced. Students with a grade of excellent or fail are fewer than those receiving other grades. If the prediction model is directly trained by the original imbalanced datasets, it will cause the problem of minority class deviation. The cross-entropy function (8) is still used to obtain the loss function between the predicted result and actual grades.

## 4. Experimental Results and Discussion

### 4.1. Setting of Experiments

#### 4.1.1. Dataset

The experimental dataset used in this research is the Open University Learning Analytics (OULA) dataset [40], which is one of the most well-known open and high-quality datasets in the field of educational data mining. It includes demographic features of students and interactive records of students' learning activities in the virtual learning environment (VLE). The dataset contains the academic performance grades, 32,593 students' scores of assessments in 22 courses and 10,655,280 records of interaction between students and online learning activities.

There are four reasons we used the OULA dataset as the experimental dataset in this study. Firstly, the dataset contains logs of students' online learning activities, which is consistent with the goals of this study. Secondly, the dataset contains the performance grade of each student in each course, which can be used to evaluate learning. Thirdly, the volume of the dataset is extremely large, which can effectively avoid overfitting during model training and ensure that the model has a better generalization ability. Fourthly, the dataset is publicly available, and many studies have been conducted on it, which allows for meaningful comparison.

In this dataset, the performance of students in each course is described by four grades: Distinction, Fail, Pass, and Withdraw. As shown in Figure 4, in 22 courses, the number of students with different grades is imbalanced, and the students obtaining a Fail or Distinction are always fewer than the students obtaining a Pass and Withdraw.

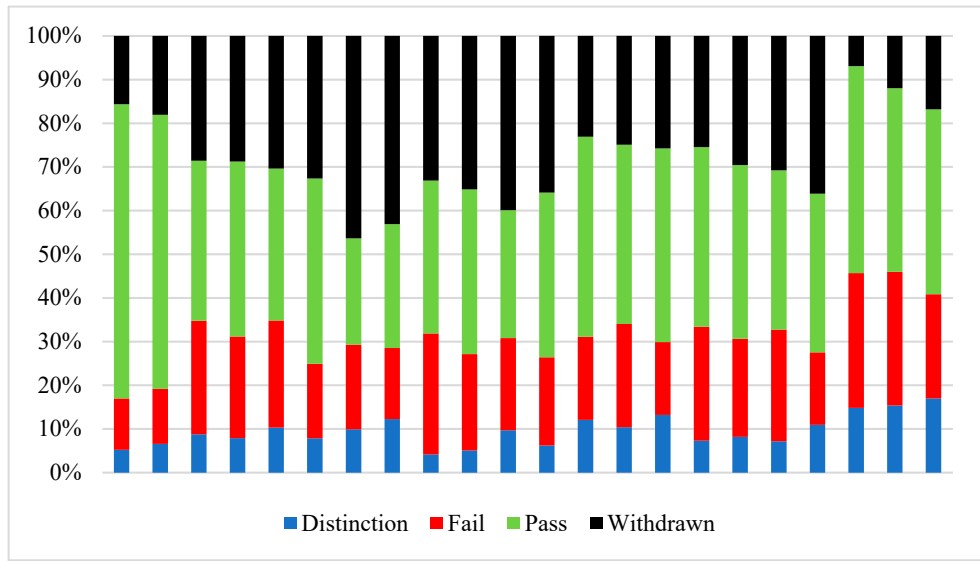

**Figure 4.** Proportion of students with different performance in the dataset.

4.1.2. Experimental Environment and Parameters of Training

We used Numpy, Pandas, and Pytorch, which are famous Python packages to represent, normalize, and construct autoencoder and prediction models in practice. We performed the experimental program on a high-performance computer. The fundamental configuration of this computer is i7-4790 3.6 Ghz, 4 core, 16 GB memory, NVIDIA geforce GTX 1080 Ti. The five baseline models used for comparison were from scikit-learn 1.2.1, which are LogisticRegression, RandomForestClassifier, LinearSVC, KNighborsClassifier and MultinomialNB. These models are most commonly used by researchers to predict students' performance [2]. The parameters of these models were the default values provided by scikit-learn.

There are several critical parameters in the training autoencoder and prediction model. According to the configuration of the computer and the properties of the experimental dataset, we carried out many preexperiments, aiming to reduce the value of the loss function as much as possible and avoid overfitting. The selected training parameters are shown in Table 3.

**Table 3.** Training parameters selected in this study.

| Parameter | Value | Description |
|---|---|---|
| countEpoch | 500 | Epoch for training autoencoder and prediction model |
| batch_size | 200 | Size of each batch of training data |
| optimizer | Adam | Adaptive Moment Estimation, which is the most popular optimizer at present |
| learning_rate | 0.001 | Rate of updating artificial neuron parameters |
| test_percent | 0.25 | Proportion of dataset split for each course: 75% for training and 25% for evaluation |

### 4.1.3. Metrics

Since the prediction of students' performance is regarded as a classification problem, we used the metrics of classification to illustrate the performance of the proposed representation and the designed prediction model. Because the dataset is imbalanced, its accuracy cannot exactly illustrate the performance of the prediction model. As there are four student grades in the dataset, we used $TP_i, FP_i, FN_i, TN_i$ to represent True Positive (*TP*), False Positive (*FP*), False Negative (*FN*) and True Negative (*TN*) in the confusion matrix of each grade. The precision and recall of each grade are denoted by $P_i$, $R_i$ respectively. The accuracy (*Acc*), precision (*Pre*), recall (*Rec*), and F1-score (*F1*) of the prediction model are defined as (3)–(6), respectively. They have an equal impact on the performance of the prediction model, both the major grades and non-major grades.

$$Acc = \frac{1}{4}\sum_{i=1}^{4}\frac{TP_i + TN_i}{TP_i + FP_i + FN_i + TN_i} \tag{3}$$

$$Pre = \frac{1}{4}\sum_{i=1}^{4}\frac{TP_i}{TP_i + FP_i} \tag{4}$$

$$Rec = \frac{1}{4}\sum_{i=1}^{4}\frac{TP_i}{TP_i + FN_i} \tag{5}$$

$$F1 = \frac{1}{4}\sum_{i=1}^{4}\frac{2 \times P_i \times R_i}{P_i + R_i} \tag{6}$$

### 4.2. Results and Discussion

We performed three groups of experiments to evaluate the performance of the representation of the online learning activity sequence, autoencoder, and end-to-end prediction model. To avoid the influence of different courses, we split the experimental dataset into 22 subsets according to the courses, and the logs of each course were independently used for model training and evaluation.

The experimental dataset contains student logs on 20 activities. We chose the logs representing students' learning activities, including externalquiz, forumng, glossary, oucollaborate, oucontent, ouelluminate, ouwiki, questionnaire, quiz and resource, while ignoring students' logs of non-learning activities, such as returning to the home page, and clicking on folders. In order to eliminate random difference, all experiments were carried out five times, and the reported experimental results are the average of all results.

### 4.2.1. Evaluation of Representation

In order to evaluate the performance of the representation of the online learning activity sequence proposed in this study, we transformed the logs of students' interaction with VLE in each course as a three-dimensional tensor $R^{m \times n \times c}$ by day. Then, we divided the tensor $R$ of each course into 10 sub-tensors according to its second dimension, with the proportion of 10–100%; that is, each sub-tensor increases the length of the learning activities sequence by 10%. Sub-tensors with different proportions can be used as learning activity sequences of students in different learning phases. Clearly, the sequence of learning

activities is short at the beginning, but it will be longer and longer in the subsequent learning phases.

We used the FNN-based classifier in Figure 3 to evaluate the performance of prediction caused by the different proportions of sub-tensors. The experimental results are shown in Figure 5.

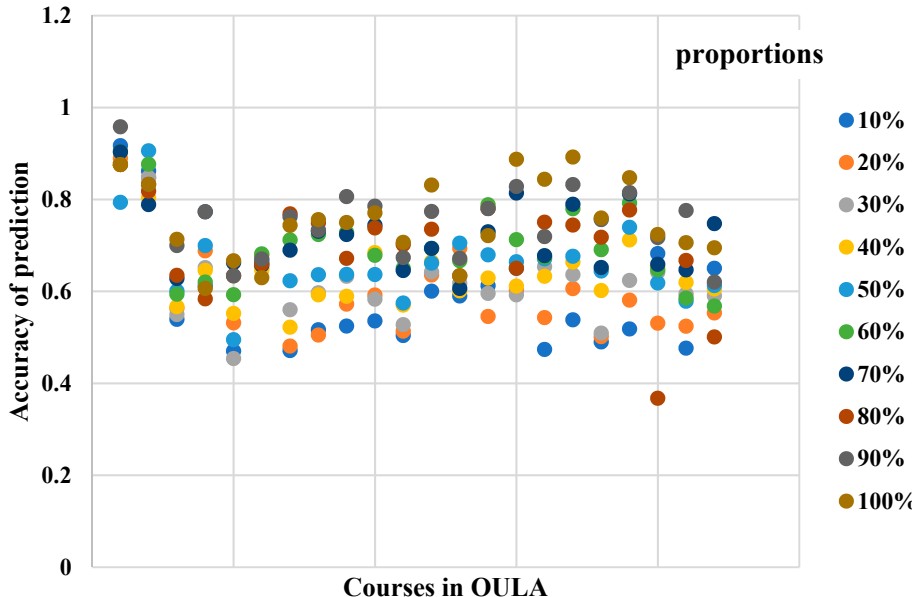

**Figure 5.** Accuracy of prediction obtained from online learning activity sequences with different proportions.

In Figure 5, there are 22 groups of data points, each of which corresponds to a course. Data points with different colors represent the prediction accuracy achieved by using online activity sequences with different proportions of this course. As can be seen from Figure 5, for most courses, a longer learning activity sequence can lead to a higher prediction accuracy, which shows that there is a strong correlation between the online learning activity sequence and the final performance of students. Moreover, 20% of sequences can make the classifier achieve an accuracy of 0.5, and 30% of sequences can achieve an accuracy of more than 0.6, which means that this representation can be used to predict learners' performance as soon as possible.

In order to compare the features representing the online learning activity sequence with other types of features, we took the online learning activity sequence, assessment scores and the demographic features of students as the inputs for the FNN-based classifier in Figure 3, used the parameters in Table 3 to train the classifier, and used 25% of the records of each course for evaluation. In the experimental dataset, the demographic features of students include gender, region, highest_education, IMD, age, previous_attempts, studied_credits, and disability. The assessment scores are the percentile scores of students' periodicity assessments in the learning process. The experimental results are shown in Table 4.

It can be seen from Table 4 that, among the three features, the students' assessment scores can achieve the best prediction results on the classifier. The prediction accuracy obtained using the learning activity sequence is 10% lower than that using assessment scores, while the prediction accuracy obtained using the students' demographic features is the lowest. Although the assessment scores achieve the best prediction results, they can only be obtained after students have completed all learning activities and cannot be used for the early and real-time prediction of performance. The prediction result of the online learning activity sequence reflects its a strong correlation with students' final academic performance, and it can be used to predict students' performance as soon as possible and

in real time. There is also a certain correlation between demographic features and students' academic performance, which can be combined with other types of features.

**Table 4.** Prediction accuracy obtained by different types of features.

| Course | Demographic Features | | | | Score of Assessment | | | | Sequence of Online Learning Activities | | | |
|---|---|---|---|---|---|---|---|---|---|---|---|---|
| | Acc | Pre | Rec | F1 | Acc | Pre | Rec | F1 | Acc | Pre | Rec | F1 |
| AAA_2013J | 0.64 | 0.16 | 0.25 | 0.20 | 0.83 | 0.41 | 0.47 | 0.43 | 0.804 | 0.204 | 0.3 | 0.24 |
| AAA_2014J | 0.61 | 0.15 | 0.25 | 0.19 | 0.74 | 0.42 | 0.41 | 0.40 | 0.768 | 0.192 | 0.3 | 0.24 |
| BBB_2013B | 0.35 | 0.25 | 0.28 | 0.26 | 0.61 | 0.42 | 0.45 | 0.42 | 0.66 | 0.444 | 0.516 | 0.456 |
| BBB_2013J | 0.37 | 0.28 | 0.28 | 0.26 | 0.67 | 0.41 | 0.42 | 0.40 | 0.564 | 0.144 | 0.3 | 0.192 |
| BBB_2014B | 0.37 | 0.32 | 0.30 | 0.29 | 0.61 | 0.42 | 0.45 | 0.42 | 0.612 | 0.492 | 0.492 | 0.456 |
| BBB_2014J | 0.44 | 0.32 | 0.31 | 0.30 | 0.69 | 0.45 | 0.47 | 0.43 | 0.576 | 0.144 | 0.3 | 0.192 |
| CCC_2014B | 0.45 | 0.14 | 0.25 | 0.17 | 0.61 | 0.38 | 0.47 | 0.38 | 0.684 | 0.42 | 0.528 | 0.432 |
| CCC_2014J | 0.39 | 0.30 | 0.28 | 0.27 | 0.61 | 0.35 | 0.46 | 0.38 | 0.696 | 0.564 | 0.552 | 0.528 |
| DDD_2013B | 0.36 | 0.27 | 0.28 | 0.27 | 0.63 | 0.43 | 0.45 | 0.44 | 0.696 | 0.468 | 0.504 | 0.492 |
| DDD_2013J | 0.42 | 0.32 | 0.31 | 0.30 | 0.67 | 0.44 | 0.49 | 0.44 | 0.708 | 0.48 | 0.516 | 0.492 |
| DDD_2014B | 0.36 | 0.27 | 0.27 | 0.25 | 0.60 | 0.40 | 0.47 | 0.40 | 0.648 | 0.468 | 0.528 | 0.492 |
| DDD_2014J | 0.43 | 0.28 | 0.30 | 0.28 | 0.69 | 0.48 | 0.51 | 0.48 | 0.768 | 0.516 | 0.552 | 0.528 |
| EEE_2013J | 0.45 | 0.14 | 0.25 | 0.18 | 0.68 | 0.37 | 0.46 | 0.39 | 0.588 | 0.144 | 0.3 | 0.192 |
| EEE_2014B | 0.33 | 0.23 | 0.26 | 0.24 | 0.54 | 0.35 | 0.38 | 0.31 | 0.66 | 0.6 | 0.588 | 0.588 |
| EEE_2014J | 0.40 | 0.21 | 0.26 | 0.21 | 0.46 | 0.31 | 0.36 | 0.27 | 0.54 | 0.132 | 0.3 | 0.192 |
| FFF_2013B | 0.39 | 0.27 | 0.28 | 0.26 | 0.66 | 0.44 | 0.47 | 0.45 | 0.78 | 0.54 | 0.576 | 0.552 |
| FFF_2013J | 0.39 | 0.28 | 0.28 | 0.27 | 0.67 | 0.45 | 0.48 | 0.45 | 0.732 | 0.504 | 0.564 | 0.516 |
| FFF_2014B | 0.35 | 0.25 | 0.27 | 0.25 | 0.65 | 0.45 | 0.48 | 0.45 | 0.516 | 0.132 | 0.3 | 0.18 |
| FFF_2014J | 0.40 | 0.31 | 0.30 | 0.29 | 0.60 | 0.39 | 0.47 | 0.41 | 0.78 | 0.528 | 0.588 | 0.552 |
| GGG_2013J | 0.42 | 0.19 | 0.24 | 0.21 | 0.70 | 0.37 | 0.42 | 0.39 | 0.576 | 0.144 | 0.3 | 0.192 |
| GGG_2014B | 0.37 | 0.22 | 0.26 | 0.23 | 0.67 | 0.37 | 0.42 | 0.38 | 0.564 | 0.348 | 0.432 | 0.384 |
| GGG_2014J | 0.36 | 0.24 | 0.26 | 0.23 | 0.66 | 0.39 | 0.41 | 0.37 | 0.636 | 0.576 | 0.564 | 0.564 |
| Average | 0.41 | 0.25 | 0.27 | 0.25 | 0.65 | 0.4 | 0.45 | 0.4 | 0.55 | 0.31 | 0.38 | 0.33 |

A comparison of the prediction accuracy achieved by the three features in 22 courses is shown in Figure 6.

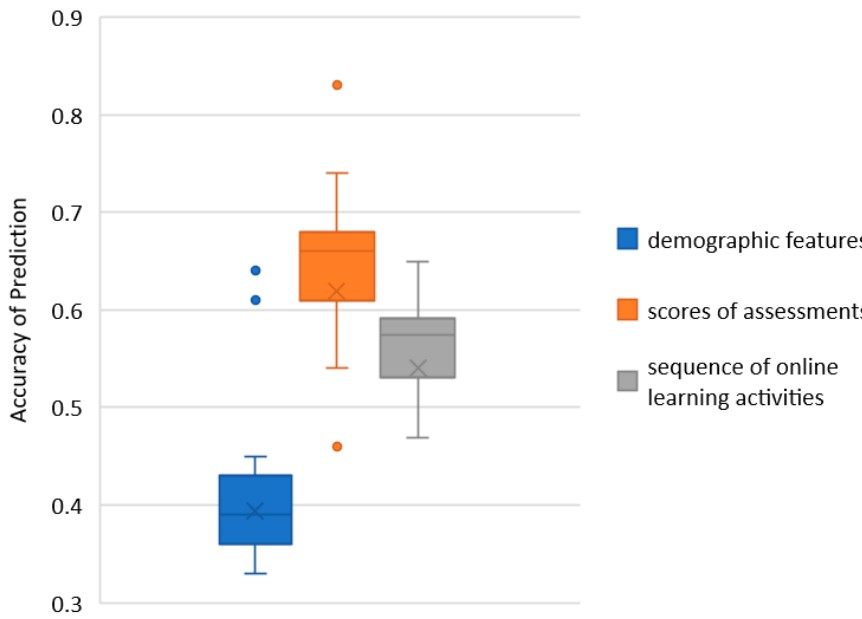

**Figure 6.** The comparison of prediction accuracy achieved by the three features.

As can be seen from Figure 6, there are two outliers when using demographic features and assessment scores, which indicates that these two types of features are occasionallt abnormally related to student performance. Therefore, using these two kinds of features probably cannot stably predict students' final performance of students. When the online

learning activity sequence is used as a feature, the prediction accuracy has no outliers, which shows that students' online learning activity sequence can be used to stably predict students' performance.

### 4.2.2. Evaluation of Autoencoder

The online learning activity sequence obtained by the proposed representation directly has the problem of high dimensions. In order to alleviate this issue, we design an autoencoder to transform the two-dimensional vector $R^{m,n \times c}$ into $R^{m,code\_size}$, and code_size should be much smaller than $n \times c$. In this group of experiments, we set code_size to 3 and used visualization to evaluate the autoencoder. The experimental dataset contains sequences of students' online learning activities over multiple semesters. In order to avoid redundancy, we only selected the results of one semester of each course to show the performance of the autoencoder. The experimental results are shown in Figure 7.

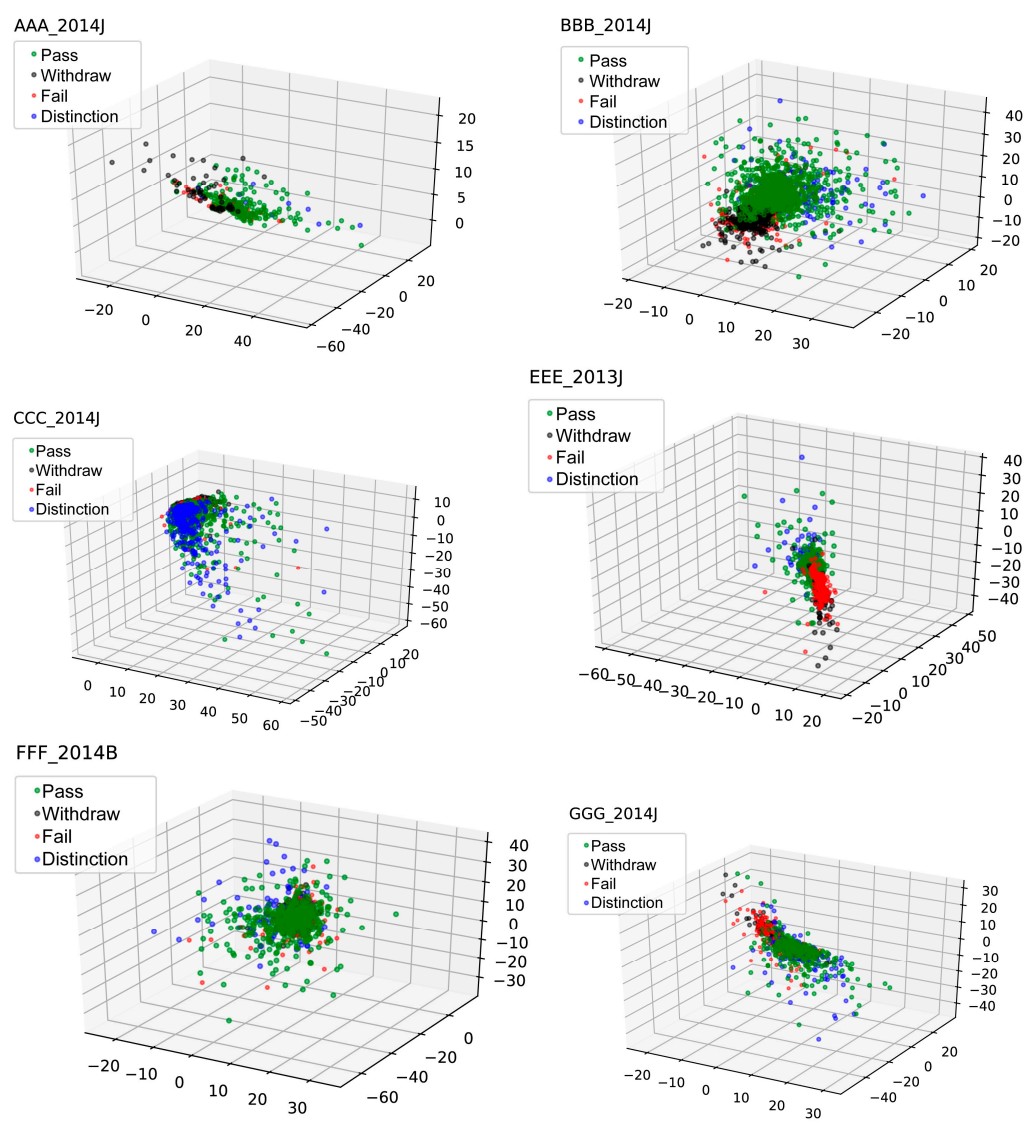

**Figure 7.** Visualization of the results of the autoencoder.

In Figure 7, each data point represents a student, and the color of the data point represents the final performance of the student. As can be seen from Figure 7, the autoencoder we designed can compress students' high-dimensional features into a smaller feature space, which is more suitable for clustering, classification, and other tasks. In Figure 7, students with the same performance in a course have a closer distance, which shows that the au-

toencoder can extract latent features that have a significant impact on performance. Since extracting latent features from the original high-dimensional features using an autoencoder is an unsupervised learning task, it also has the advantage of not requiring the special annotation of students.

In addition, we set the code_size to 6, 9, 12, 18, 24, 36, 48, respectively. The purpose of this was to evaluate the influence of latent features with different sizes on the prediction results. In order to avoid the influence of random factors, we compared the average value of the prediction results of all courses. The experimental results are shown in Figure 8.

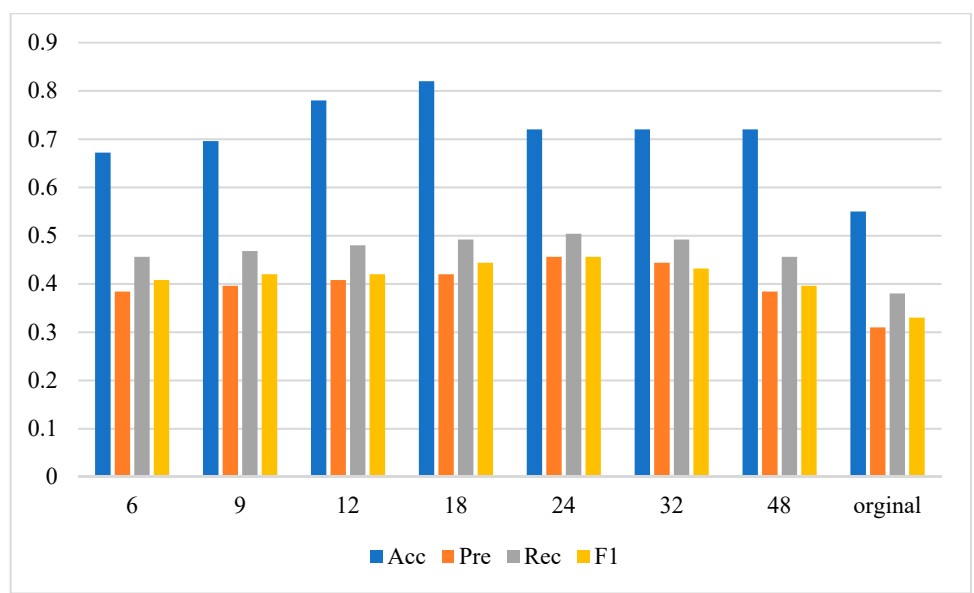

**Figure 8.** Influence of different sizes of latent features on prediction results.

As can be seen from Figure 8, the accuracy of prediction using the latent features extracted by the autoencoder generally improved; when code_size is 18, the improvement in prediction accuracy is the most significant, reaching 25%. This also shows that the autoencoder can extract more essential features, which can affect the performance of the original online learning activity sequence. However, when using latent features, the Pre, Rec, and F1 of the prediction model do not significantly improve, which indicates that the accuracy of the prediction model for non-major grades is unsatisfactory. The bias of the prediction model may be caused by an imbalanced training dataset.

### 4.2.3. Evaluation of End-to-End Prediction Model

In order to alleviate the bias caused by the imbalanced training dataset and improve the prediction accuracy of the model for non-major grades, we resampled the training dataset before training. We used an oversampling method that can make full use of the information in the training dataset. This method can synthesize a new record based on the existing records to make the dataset roughly balanced. As can be seen from Figure 7, the latent features generated by the autoencoder of students with the same grades have a closer distance in the feature space, so we used the distance-based algorithm KMeansSMOTE [41] to oversample the imbalanced training dataset. KMeansSMOTE uses K-MEANS [42] for clustering and SMOTE [43] for oversampling, including three steps, clustering, filtering, and oversampling, which avoids the generation of noise and effectively overcomes the imbalance between and within classes. The end-to-end prediction model composed of the pretrained autoencoder and the classifier trained with the resampled dataset is shown in Figure 5, and the results of its evaluation are shown in Figure 9.

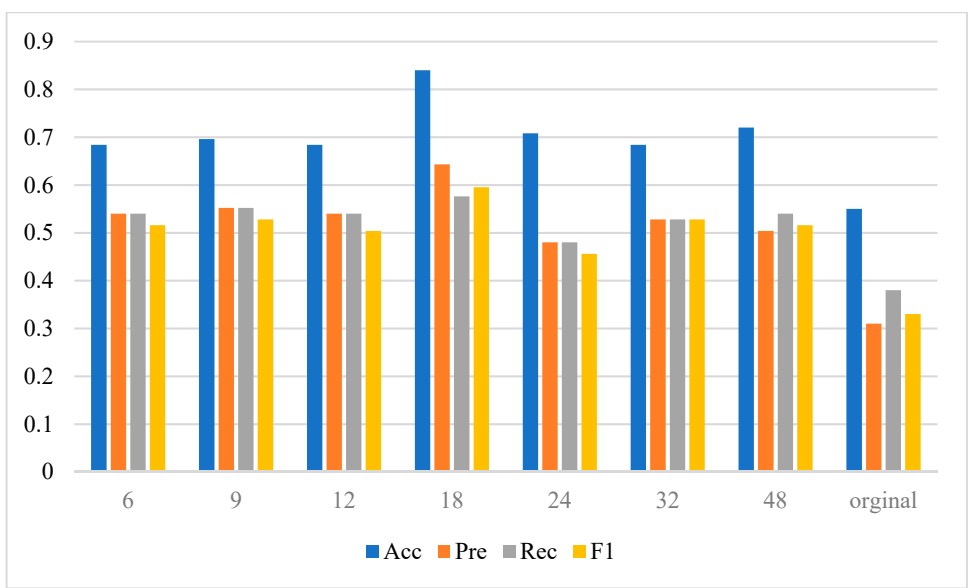

**Figure 9.** Evaluation of end-to-end prediction model.

It can be seen from Figure 9 that the new prediction model we designed has a better prediction performance than the original model, especially in the three metrics of Pre, Rec, and F1, which increased by more than 10%, reflecting that the end-to-end prediction model has a better prediction accuracy for non-major grades. In particular, when code_size is 18, the new model has the best performance. Clearly, the size of the latent features extracted by the autoencoder has a significant impact on the prediction performance, and how to set the size of the latent features is a problem that needs further discussion.

We compared the performance of our model with five selected baseline models, and the results are shown in Table 5. From Table 5, we can see that our model has obvious advantages compared to the five baseline models, especially for non-major academic grades.

**Table 5.** Performance comparison between our model and baseline model.

| Model | Acc | Pre | Rec | F1 |
|---|---|---|---|---|
| Naïve Bayes | 0.47 | 0.14 | 0.26 | 0.17 |
| SVM | 0.58 | 0.41 | 0.39 | 0.35 |
| Logistic Regression | 0.53 | 0.29 | 0.32 | 0.25 |
| Random Forest | 0.66 | 0.56 | 0.48 | 0.46 |
| KNN | 0.38 | 0.34 | 0.32 | 0.27 |
| Our Model | 0.84 | 0.64 | 0.57 | 0.59 |

## 5. Key Findings and Future Research

### 5.1. Key Findings

(1) Students' online learning activity sequences can be used to effectively predict students' learning performance. Compared with students' demographic features and assessment scores, a small part of the online learning activity sequences can be used to predict students' performance at the beginning, rather than waiting for all learning activities to be completed. The prediction result based on online learning activity sequence has better stability, and the prediction accuracy is proportional to the length of the online learning activity sequence.

(2) The autoencoder based on a deep neural network can extract latent features with lower dimensions from the original high-dimensional online learning activity sequences, which contains the essential information of learners' online learning behavior, so that

students with the same performance have a closer distance in the new feature space. Using latent features can further improve the accuracy of performance prediction.

(3) The classifier based on a deep feedforward neural network can be used to predict students' performance. Although there is no clear standard for the selection of hyperparameters of this network, the experimental results show that the network with more than five hidden layers and a single hidden layer containing up to 512 artificial neurons can achieve a prediction accuracy of more than 70%. The parameters used to train the autoencoder and classifier based on a deep neural network need to be selected through experiments according to the hardware, volume and properties of the training datasets.

(4) The training dataset containing students' performance is often imbalanced, which leads to bias in the prediction model for non-major grades. Distance-based and oversampling methods such as KMeansSMOTE can generate a new, balanced training dataset and improve the performance of the prediction model.

### 5.2. Future Research

Aiming to predict students' performance, which is an important problem in educational data mining, this study proposed a representation of students' online learning activity sequence and designed an autoencoder and end-to-end performance prediction model based on a deep neural network. Important problems that need to be studied in the future include the joint representation of various types of features, the continuous improvement in the autoencoder to enhance the representativeness of extracted latent features, and the method of setting hyperparameters and training parameters for a deep neural network.

**Author Contributions:** Conceptualization, methodology, writing—original draft preparation, X.W.; software, validation writing—review and editing, H.J. All authors have read and agreed to the published version of the manuscript.

**Funding:** This work was supported by Anhui Philosophy and Social Sciences Planning Youth Project [Grant Number AHSKQ2022D097].

**Data Availability Statement:** The OULA dataset is freely available at https://analyse.kmi.open.ac.uk/open_dataset (accessed on 1 December 2022) under a CC-BY 4.0 license.

**Conflicts of Interest:** The authors declare no conflict of interest.

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
