# Peer review of "Early Prediction of Students’ Performance Using a Deep Neural Network Based on Online Learning Activity Sequence"

_applsci, doi:10.3390/app13158933_

Round 1

Reviewer 1 Report

Dear authors,

Congratulations on your work, I'd like to share some notes, in order to contribute to the text.

Text problems:

Pag. 5

Table 1 is cut on Duration(Minute) column

As can be seen from Figure 1(a).  there's no subfig (a) marked 

pag 9.

Table 2.  is cut too.

pag 10.

fig 5. it is hard to understand what the numbers on the right mean.

pag11.

Table 3. could have  vertical lines separating the three features group.

pag 12.

Figure 6.  is presenting  Chinese words  as  the vertical labels

Suggested improvements:

In the section Related Work, you could summarize the state of the art as a table.

Author Response

Dear Reviewer:

Based on your valuable comments, the manuscript has been revised. Our revisions or responses to each comment are as follows:

Text problems:

  • 5 Table 1 is cut on Duration(Minute) column

Reply: This issue is caused by the automatic typesetting of submission system, and the Duration(Minute) is the last column in Table 1.

  • As can be seen from Figure 1(a). there's no subfig (a) marked

Reply: This issue is caused by the automatic typesetting of submission system, and Figure 1(a) is the left sub fig in Figure 1.

  • pag 9.Table 2. is cut too.

Reply: This issue is caused by the automatic typesetting of submission system, and the description is the last column in Table 2.

  • pag 10.fig 5. it is hard to understand what the numbers on the right mean.

Reply: The numbers on the right of Figure 5 represent different proportions of the activities sequence. We have revised the labels for the x and y axis in Figure 5 and added labels to the numbers on the right side as percentages, making the meaning of Figure 5 clearer.

  • Table 3. could have vertical lines separating the three features group.

Reply: We have added vertical lines for separating three group of features in Table 3

  • pag 12.Figure 6. is presenting Chinese words  as  the vertical labels

Reply: This issue is caused by the automatic typesetting of submission system, and we have revised Figure 6.

Suggested improvements:

In the section Related Work, you could summarize the state of the art as a table.

Reply: We have added Table 1 for comparing related work

Reviewer 2 Report

The proposed paper describes the possible use of artificial intelligence systems in education and especially within mooc-type digital learning environments. The paper is clear in its experimental definition, the data are correct and clear. I have analyzed the code and the expert is well structured and the use of encoders and neural networks is clear. Since the subject is also of interest to teachers, one could think of including some cross-references in the references to help the less expert reader.

Author Response

Dear Reviewer:

Based on your valuable comments, the manuscript has been revised. Our revisions or responses to each comment are as follows:

(1)The proposed paper describes the possible use of artificial intelligence systems in education and especially within mooc-type digital learning environments. The paper is clear in its experimental definition, the data are correct and clear. I have analyzed the code and the expert is well structured and the use of encoders and neural networks is clear. Since the subject is also of interest to teachers, one could think of including some cross-references in the references to help the less expert reader.

Reply: We have added four references in the references, which helps readers better understand this manuscript.

  1. Xiao, W. and J.J.E.R. Hu, A state‐of‐the‐art survey of predicting students' performance using artificial neural networks. 2023: p. e12652.
  2. Yang, Z., et al., Autoencoder-based representation learning and its application in intelligent fault diagnosis: A review. 2022. 189: p. 110460.
  3. Hastie, T., et al., Overview of supervised learning. 2009: p. 9-41.
  4. Ericsson, L., et al., Self-supervised representation learning: Introduction, advances, and challenges. 2022. 39(3): p. 42-62.

Reviewer 3 Report

+  significant problem

+ improvement compared to the baseline approaches

- The problem statement did not become clear to me rendering the significance and novelty challenging to assess.

- The link from the approach to the evaluation results is broken.

A professional language editing service is strongly recommended for several bad English constructions, grammar mistakes, misuse of articles, and problems with hyphenation.

A professional language editing service is strongly recommended for several bad English constructions, grammar mistakes, misuse of articles, and problems with hyphenation.

Author Response

Dear Reviewer:

Based on your valuable comments, the manuscript has been revised. Our revisions or responses to each comment are as follows:

  • improvement compared to the baseline approaches

Reply: We have added the comparison of related works in subSection II. B and compared our model with five baseline models in Table 5.

(2) The problem statement did not become clear to me rendering the significance and novelty challenging to assess.

Reply: We added the definition of early prediction of students’ performance in subsection II.A. Additionally, We have revised the paragraphs following table 1. The contribution of this study is highlighted by illustrating issues that were not concerned in previous studies

(3) The link from the approach to the evaluation results is broken.

Reply: The representation method of online learning activity sequence , the autoencoder used to extract latent features from the sequence,the end-to-end early prediction model of students’ performance is proposed in Section III. The evaluation results of the above methods and models are reported in Section IV. B respectively. In order to clarify the link from the approach to the evaluation results, we have added some descriptions of novelty in the Section III.

(4) A professional language editing service is strongly recommended for several bad English constructions, grammar mistakes, misuse of articles, and problems with hyphenation.

Reply: Due to the regulations of the author's organization, we are unable to use professional language editing services. We carefully reviewed the manuscript and revised bad English constructions and grammar mistake as much as possible.

Reviewer 4 Report

Based on the observations, I have the following comments:

---Abstract is ambiguous and not clear about the novelty.

---Related work is way too theoretical and does not contribute much to findings. 

---Prediction accuracy is relatively very low as per the practical scenarios of the proposed model.

---Basis for metric selections was not given.

---Learning measures and different standards were not considered within datasets.

---. Require more simulations over different datasets which should consider the age, backlogs, health issues, etc.

Kindly go with proofread 

Author Response

(The authors gave the same response as above.)
